# Role of Bacteriophages for Optimized Health and Production of Poultry

**DOI:** 10.3390/ani12233378

**Published:** 2022-12-01

**Authors:** Rao Zahid Abbas, Abdullah F Alsayeqh, Amjad Islam Aqib

**Affiliations:** 1Department of Parasitology, University of Agriculture, Faisalabad 38000, Pakistan; 2Department of Veterinary Medicine, College of Agriculture and Veterinary Medicine, Qassim University, Buraidah 51452, Saudi Arabia; 3Department of Medicine, Cholistan University of Veterinary and Animal Sciences, Bahawalpur 63100, Pakistan

**Keywords:** bacteriophages, poultry, health, production, drug resistance, biosecurity and biosafety

## Abstract

**Simple Summary:**

The poultry industry has emerged as a significant contributor in meeting the requirement for animal meat. Meanwhile, there arose health issues that were dealt with from time to time with various treatment approaches, such as the use of antibiotics, prebiotics, and probiotics that specifically fight against pathogens. It is very important to find more refined alternative therapeutics to tackle threats to the health and production of poultry. Bacteriophages have emerged as effective alternatives to antibiotics to tackle pathogens with minimized adverse effects. They can be used to target pathogens alone or pathogens in groups. The studies regarding bacteriophages in poultry production require finding a concise review of their mechanism of action: The required questions are like what are the benefits of poultry production? What are the threats associated with it? What is the role of bacteriophages in biosecurity and biosafety? And what is their impact on the economic situation? This review thus covers these aspects to come up with a better understanding of the use of bacteriophages in the poultry industry. This review is expected to open horizons to further research work on bacteriophages for poultry and to promote product development to lower the burden of antimicrobial resistance.

**Abstract:**

The poultry sector is facing infections from *Salmonella*, *Campylobacter*, *Listeria* and *Staphylococcus spp*., and *Escherichia coli*, that have developed multidrug resistance aptitude. Antibiotics cause disturbances in the balance of normal microbiota leading to dysbiosis, immunosuppression, and the development of secondary infections. Bacteriophages have been reported to lower the colonization of *Salmonella* and *Campylobacter* in poultry. The specificity of bacteriophages is greater than that of antibiotics and can be used as a cocktail for enhanced antibacterial activity. Specie-specific phages have been prepared, e.g., Staphylophage (used against *Staphylococcus* bacteria) that specifically eliminate bacterial pathogens. Bacteriophage products, e.g., BacWash^TM^ and Ecolicide PX have been developed as antiseptics and disinfectants for effective biosecurity and biosafety measures. The success of phage therapy is influenced by time to use, the amount used, the delivery mechanism, and combination therapy with other therapeutics. It is a need of time to build a comprehensive understanding of the use of bacteriophages in poultry production. The current review thus focuses on mechanisms of bacteriophages against poultry pathogens, their applications in various therapeutics, impacts on the economy, and current challenges.

## 1. Introduction

The poultry industry has emerged as a robust widespread sector at the world level, producing 133,266 thousand tons of carcass-weight of poultry meat, as per a 2020 FAO report [1]. On the other hand, poultry pathogens inclusive of which are strains from *Campylobacter*, *Salmonella*, *Staphylococcus*, *Escherichia*, and *Listeria* are of serious concern for poultry health and production, as read by the European Centre for Disease Prevention and Control, 2019 [2]. The intention to maximize production by culminating infection through antibiotics is leading towards the conversion of multiple drug resistance to pan drug resistance [3]. The use of antibiotics in poultry feed has significantly increased the development of resistance [4] that ultimately circulates at tiers of animals, humans, and the environment [5].

The bacteriophages are viruses that specifically infect and kill bacteria and archaea. These are divided into several orders and nearly 15 families, while most of these phages belong to Caudovirales with broad ranges of isometric heads of 20–200 nm, thus making them 1000 times smaller than the average bacteria [6]. Their activity has been found promising both against Gram-positive and Gram-negative bacteria. They also attack selective bacteria thus proceeding further for the selective elimination of bacterial pathogens [7]. Thus, keeping in mind the emerging resistance in bacteria of poultry and threats to public health, the current review was intended to cover mechanisms of phages as therapeutics, their usage in the industry, and limitations/threats involved with the usage of bacteriophages. Most studies have focused on how effectively bacteriophages can lower bacterial counts and manage zoonotic bacterial infections in poultry, which significantly affect public health [8,9,10]. The current review thus covers every aspect of bacteriophages in poultry health and production.

## 2. Introduction to Bacteriophages in Biomedicine

### 2.1. History

The first communication about the discovery of a whole new world of small organisms’ dates to 1674 when the talented self-proclaimed Dutch optician Antoine von Leeuwenhoek reported his observations in some of his earlier letters to the Royal Society of London. Microbiology as a separate branch of biological science, however, emerged only in the second half of the eleventh century with its methodology and theoretical concepts, mainly due to the important work of Louis Pasteur that laid the foundation for the main branches of modern microbiological science such as physiology and biochemistry of micro-organisms, industrial microbiology, medical microbiology, and vaccinology. However, the ultimate triumph of bacteriological methods in medicine and other disciplines has only been possible due to the research of German microbiologist Robert Koch. Together with his colleagues, Koch formulated the basic principles of modern methodology for working with bacterial cultures, which are still relevant today. Microbiologists from Koch and Pasteur’s schools, as well as their followers around the world, isolated and described more than 25 agents of a bacterial infection within a short period from 1876–1912. At the same time, the work of the Pasteur School and the first immunologist resulted in the development of vaccines against a series of very dangerous infections, as well as toxic neutral sera such as anti-diphtheria serum and anti-plague serum. The bacteriological assessment was introduced into clinical practice. Specialized laboratories have been arranged to produce medicinal sera and vaccines, which require the production of large amounts of bacterial biomass to be isolated, for example, enough related toxins for animal immunization [11]. Bacteriophages were independently discovered in 1915 by Frederick Twort, a British pathologist, and discovered in 1917 by French-Canadian microbiologist Felix de Herelle. Twort described the “glass change” of micrococci colonies, while De Herelle isolated the “antimicrobe” of Shigella and coined the term ‘bacteriophage’—which means bacterial eater. Phages bind to the internal parasites of bacteria and have diverse life cycles. The ability of phages to infect bacteria led De herelle to test their therapeutic potential against bacterial infections. Even in their first paper, they noted that the presence of phages correlated with disease clearance in dysentery patients, and they studied rabbits in which phages were protected from Shigella infection. Much of the early phage research conducted in the 1920s and 1930s focused on the development of phage therapies for the treatment of bacterial infections, and companies began to market phage preparations. However, in the late 1930s, the American Medical Association’s Council on Pharmacy and Chemistry concluded that the efficacy of phage therapy was ambiguous, and more research was needed. In this timeline, we highlight the effects of phages in the first 100 years since their discovery in terms of the origins, ecology, and evolution of molecular biology, and their biotechnological exploitation [10].

### 2.2. Applications

Bacteriophages are prevalent in every environment and are considered approximately 10 times the number of known bacteria making them solid candidates to eliminate infectious diseases [11]. The commensal gut flora is not destroyed by this mode of action. During therapy, bacteriophages self-replicate, thus they do not need to be applied frequently. The inability of phages to attach to and multiply in eukaryotic cells results in a decline in their titer, which is associated with a significantly lower number of pathogenic bacteria causing an infection in the organism. Another essential characteristic is that phages are non-toxic since they are often made up mostly of proteins and nucleic acids [12]. The increasing usage of bacteriophage therapy is due to the phages’ great specificity for a given bacterial species as well as their capability to lyse infected bacteria and mutate resistant bacteria. Hospital-acquired bacteria are multi-drug resistant which makes them a major cause of resistance spread among individuals outside the hospitals. Phage therapy is used to combat bacterial infections in the fields of dermatology, stomatology, otolaryngology, ophthalmology, gynecology, pediatrics, gastrointestinal, urology, and pulmonology [13]. Approximately 85% of human infections treated with bacteriophages are successful, especially those caused by combined infections primarily caused by *Staphylococcus aureus* (*S. aureus*), *Klebsiella*, *E. coli*, *Proteus*, *Pseudomonas*, *Enterobacter*, and Vancomycin-resistant *Enterococci* [14]. Antibiotics are considerably less specific than bacteriophages, as the latter can infect species, serotypes, and strains. In addition to destroying pathogenic bacteria, antibiotics also alter the microbiota of the gut, increasing the risk of dysbiosis, immunosuppression, and secondary infections. As a result, novel bacteriophage therapies are highly effective in treating bacterial infections in poultry [15].

## 3. Experimental Studies of Bacteriophages in Poultry

### 3.1. Campylobacter spp.

*Campylobacter* spp. is found everywhere and prefers the bird’s gut where they live as commensals. Poultry has evolved into a natural reservoir for *Campylobacter* spp. due to its favorable (optimum) body temperature, which is the main cause of human infections [16]. Recent research has demonstrated the effectiveness of phage therapy in lowering *Campylobacter* colonization in poultry and so reducing the danger of it getting into the food chain. Broiler chickens infected with *Campylobacter jejuni* (*C. jejuni*) were given an oral dose of a phage cocktail, including virulent *Campylobacter* phages [17]. Bacteriophages specifically decreased the prevalence of *C. jejuni* but microbiota did not reduce. According to research, reducing the amount of *C. jejuni* in chickens using bacteriophage control might decrease human exposure and morbidity caused by eating infected poultry products [18] (Figure 1).

### 3.2. Salmonella spp.

*Salmonella* is the second-most important zoonotic foodborne pathogen (after *Campylobacter*) and a major bacterium that affects commercial poultry. In the early 1990s, Berchieri et al. [19] demonstrated the effectiveness of bacteriophages by simultaneously administering phages and *S. typhimurium* to chickens. During the trial, bacteriophages decreased the viable levels of *Salmonella* in the chicken’s gut (crop, small intestine, and cecum). More than one log_10_ reduction was observed in the crop and small intestine shortly after administration and in the digestive tract three days later. Although certain phages (AB2) multiplied, they had no impact on the level of *Salmonella* present in the cecum. The chicken serum did not contain any phage-neutralizing antibodies (at 32 dpi). Phages could also be discriminated against by interacting with birds. Additional studies supported the finding that a cocktail of bacteriophages made up of many phages was better than a single phage in inducing the lysis of *Salmonella* spp. Additionally, the cocktail made from *S. enteritidis,* and *S. typhimurium* could stimulate the lysis of different *Salmonella* serovars (Virchow, Hadar, Infantis). *Salmonella* cell counts in chicken cecum appear to have significantly decreased after recurrent oral administration of the bacteriophage cocktail [20]. *Salmonella*-specific bacteriophages were found in poultry sewage samples and diseased broiler chickens, according to other studies. *Salmonella* infections were treated with bacterial phages taken from sewage water before being given to chicks orally, followed by four more phage therapies. After the fifth dosage, no pathogen (bacteria) was found in the cecum, revealing that phage treatment for *Salmonella* had successfully cured the chickens (at 15 dpi [21]. Using *S. enteritidis*, other authors have revealed the whole sequence of the genome of a bacteriophage that was separate from water. Both *S. enteritidis* and *C. jejuni* may be infected by the bacteriophage, which can be applied to poultry as phage treatment and to enhance the biosafety of poultry meat (Figure 2).

### 3.3. Escherichia spp.

*E. coli* is a Gram-negative bacterium that often lives in birds’ digestive tracts and is widely distributed through feces. This assertive microbe can cause infections that are both primary and secondary. All ages and types of poultry are frequently affected by *E. coli*-associated infections. Bacteriophage, which was initially produced from the sewage of humans, was found in research by Barrow et al. [22] to be successful in preventing and treating septicemia, cerebritis, and meningitis in chickens. *E. coli* was delivered intramuscularly or intracranially to chickens, while phage preparations were injected intramuscularly (gastrocnemius muscle, right leg). The non-treated 3-week-old and newly born chicks that were injected using both methods had a mortality rate of approximately 100%. The chicken intracranially infected with *E. coli* were found to have the phages reaching their brain. They might grow quickly and reduce the number of bacteria. Furthermore, the above authors revealed that bacteriophages can protect chickens even when the treatment is given (one–two days) in anticipation of an *E. coli* challenge or when clinical signs first appear. This might suggest that phages can stay in the tissues for a sufficient period and can thus be utilized to prevent colibacillosis as well as treat it. Intriguingly, the authors also found that VirkonTM (Antec International, Sudbury, United Kingdom), a widely employed virucidal disinfectant, was quite effective against the phage used. These discoveries, however, need to be proved in future studies [20]. By spraying bacteriophages onto 7-day-old chickens before exposing them to three different *E. coli* challenges, Huff et al. [23] have shown that it is possible to avoid the development of airsacculitis caused by *E. coli*. However, when given to the chicken after it had been exposed to *E. coli*, the bacteriophage aerosol spray proved ineffective. The circulating bacteriophage titers appear to have an impact on how well bacteriophage-based treatments work. Only a few chickens were found to have measurable levels of bacteriophage in the blood, in contrast to the intramuscular injection of bacteriophages. Other findings showed that bacteriophage therapy was equivalent to enrofloxacin treatment. Additionally, the synergistic effects of bacteriophage and enrofloxacin when administered together increased the effectiveness of colibacillosis therapies. Among several bacteriophages are podoviruses which are exclusively lytic viruses [24] and which, as others, affect a wider range of bacteria, such as *S. aureus* [25], *E. coli*, and others. The number of antibiotics required to treat bacterial diseases may be reduced if the antibiotic is combined with bacteriophage treatment [20].

### 3.4. Clostridium spp.

In the natural ecosystem, *Clostridium perfringens* (*C. perfringens*) is widely distributed and is a typical component of the gut microbiota of chickens. It is non-pathogenic at low population levels (<10^4^ CFU), however, most of its morbidity is linked to germs. As a result of the discovery of bacteria derived from the *C. perfringens* strain in soil, sewers, and drainage water of poultry processing facilities, the order Caudovirales was recognized as having members of the families Siphoviridae and Podoviridae. The phages were unable to infect several different *C. perfringens* strains. Furthermore, phage activity appeared to be limited to a strain of this bacterium. Endolysin encoded by *C. perfringens* phages, according to some authors, may be particularly helpful for controlling this pathogen. However, there may be differences in the sensitivity of the various strains, the results suggested that endolysin may be effective against all the *C. perfringens* strains examined. Bacteriophage (INT-401) is effective in preventing necrotic enteritis (NE), which is caused by *C. perfringens* in poultry. The phage therapy through food or water permitted the experimentally infected broiler chicken to gain weight more quickly and to have a higher conversion of feed ratio (FCR), in addition to lowering the mortality rate [20] (Figure 3).

## 4. Effect of Bacteriophages on Poultry Immunity against the Challenge of Salient Bacteria

The threat of pathogens has become increasingly prominent in public health. In poultry, *Salmonella enterica* subspecies *enterica* serovar *Enteritidis* (*S. enteritidis*), *Salmonella enterica* subspecies *enterica* serovar *typhimurium* (*S. typhimurium*), *Escherichia coli*, *Listeria monocytogenes*, and methicillin-resistant *Staphylococcus aureus* can be found. Bacteriophages have been studied mainly for their efficacy in reducing bacterial counts and controlling bacterial infections in poultry, which are zoonotic and have significant public health consequences [8,26]. The European Food Safety Authority (EFSA) and the European Centre for Disease Prevention and Control (ECDC) published recent reports (2019) that the most prevalent zoonoses in Europe (EU) are campylobacteriosis, salmonellosis, Shiga toxin-producing *E. coli*, and yersiniosis. Since lytic bacteriophages are limited in their ability to kill bacteria, they are only suitable for treating bacterial infections with phage therapy. The specificity of bacteriophages is much greater than that of antibiotics. In addition to killing pathogenic bacteria, antibiotics also alter the normal microbiota of the gastrointestinal tract, causing dysbiosis, immunosuppression, and therefore secondary infections [15].

### 4.1. Campylobacters

The bacterium *Campylobacter* spp. is ubiquitous in the environment; however, it prefers to colonize the guts of birds as a commensal. As a result of the ideal body temperature for *Campylobacter* spp., poultry is a natural reservoir for bacteria, accounting for most human infections. It takes approximately seven days after hatching for *Campylobacter* spp. to colonize chickens in poultry farms. There is usually no clinical sign or lesion associated with *Campylobacter* infection in chickens. It has been reported that *Campylobacter* spp. are present in poultry flocks in varying numbers from 2% to 100% [27]. In some studies, 91.5% of carcass surface swabs and 100% of small intestinal samples showed *Campylobacter* contamination at slaughter [28]. It was noted by other authors that broiler chickens had a lower prevalence (34.3%, cecum samples) of campylobacter.

A great deal of effort is being put into reducing *Campylobacter* because of reports about antibiotic resistance (to fluoroquinolones, tetracyclines, erythromycin, and gentamicin) as well as virulence. There is a high prevalence of bacteriophages that are specific for *Campylobacter* in commercial chickens and retail chicken products, mostly those that are related to the Myoviridae family and comparatively few that are related to the Siphoviridae family [29]. Several studies have demonstrated that phage treatment can reduce *Campylobacter* colonization in chickens, thereby minimizing the risk of bacteria entering food systems. Oral administration of virulent *Campylobacter* phages was used to treat broiler chickens colonized with *C. jejuni*. Despite reducing the abundance of *C. jejuni*, the microbiota was not affected by the bacteriophage predation of *C. jejuni*. Using bacteriophages to control *C. jejuni* levels in chickens could reduce human exposure and disease caused by contaminated poultry products [18].

### 4.2. E. coli

Many bacteria affect commercial poultry, but *Salmonella* is the second most important foodborne pathogen (after *Campylobacter).* (3) There are three types of nonmotile *Salmonella* infections: (1) A host-specific *Salmonella enterica* infection is caused by the subspecies *enterica* serovar *pullorum* (*S. pullorum*). Secondly, *Salmonella enterica* infections are caused by *S. gallinarum* (subspecies *enterica*). A disease caused by *S. pullorum* that affects young birds is pullorum disease (PD), an acute systemic infection. Asymptomatic carriers are usually infected, adults. A septicemic disease primarily affecting growing and mature birds, *S. gallinarum* causes fowl typhoid. A motile *Salmonella enterica* serotype known as *paratyphoid salmonella* (PT) is responsible for nonhost-specific infections caused by *S. typhimurium* and *S. enteritidis*. Acute systemic disease is only prevalent in young, highly susceptible birds under stressful circumstances, despite the prevalence of PT infections in poultry. It is usually found in young birds (less than four weeks old) that these signs appear. As a result of PT infections in poultry, unaffected intestinal tracts and internal organs can be colonized asymptomatically and persistently, potentially contaminating finished products. *Salmonella enterica* subspecies *arizonae* is responsible for avian arizonosis (AA), a chronic or acute infection of birds. Clinical symptoms usually do not appear until an older bird has developed the disease, but the disease can still be present in the bird [30].

### 4.3. S. aureus

The most prevalent and pathogenic staphylococcal species found in poultry isolates is *S. aureus*. A staphylophage is a phage that targets *Staphylococcus* bacteria. Staphylophage were divided into three groups (Figure 4) according to the size of their genomes: polyviridae, -class I, which has the smallest genome, In the Siphoviridae, there is an intervening genome size, while in the Myoviridae, there is a large genome [24]. *S aureus* bacteria produced in broiler chickens and turkeys led to the development of bacterial phages in the family Siphoviridae of the Caudovirales order. They had high lytic characteristics against *Staphylococcus* strains as well as other bacteria and belonged to the three serogroups (A, B, and F with the Fa, Fb subgroups). Even though these bacteriophages exhibited great *S. aureus* selectivity, some of them had enterotoxigenic genes, making them ineffective for phage therapy [25]. Myoviruses are regarded as the most promising staphylococcal phages from a therapeutic perspective. Podoviruses are exclusively lytic viruses, however, they are rarely present [24]. There are no phage preparations available right now that are designed to prevent and cure infections caused by *S. aureus* in poultry [20] (Figure 5). There are several different types of *Staphylococci* isolated from poultry, but *Staphylococcus aureus* is the most common. In poultry environments, *Staphylococci* are ubiquitous in the skin and mucous membranes of healthy birds. *Staphylococcal* infections caused by *S aureus* infect a significant number of chickens and turkeys [31]. By reducing production results and condemning carcasses at slaughter, these infections cause economic losses. It has been reported that arthritis, synovitis, chondronecrosis, osteomyelitis, gangrenous dermatitis, subdermal abscesses (bumblefoot), and green liver-osteomyelitis complexes are all attributed to *S aureus* infection [32]. Some cases of food poisoning result from the presence of enterotoxin-producing bacteria. A poultry-associated food poisoning can occur when poultry carcasses are contaminated with *S aureus* during processing (especially when enterotoxin-producing strains are present). Methicillin-resistant *S aureus* (MRSA) has also been detected in poultry meat [33].

## 5. Products of Bacteriophages Used as Vaccines or Adjuvants

The first vaccine to be investigated was a whole-cell vaccine. Bacteria are administered that have been killed or attenuated so that they are no longer virulent or colonizing. Vaccinations against *Campylobacter* are described only in the following paragraphs. In vaccinated groups compared to unvaccinated groups, caecal *C. jejuni* loads were reduced by 16–93% after vaccination with formalin-inactivated strain F1BCB. In vaccinated birds, serum or bile IgA levels were generally higher than in control birds, with more immune-responsive birds. This study did not find that heat-labile toxin (LT) adjuvant impacted vaccine efficacy. Other researchers, however, did not see similar results. In one study [34] it was demonstrated that vaccination with formalin-inactivated *C. jejuni* and complete Freund’s adjuvant produced specific antibodies in chicken serum but did not significantly affect intestinal colonization after a homologous or heterologous challenge. All birds colonized similarly to the positive control group, regardless of experimental conditions, even when vaccinated with four viable but noncolonizing strains of *C. jejuni* [35]. WCV was tested with purified flagellin instead of alone, according to Widders et al. [36]. The second vaccination only reduced caecal *C. jejuni* loads when it was administered intraperitoneally and not orally. After immunization in ovo and oral boosters after hatching, chicks with serum IgY, IgA, and IgM, as well as bile IgA, showed a strong immune response. A higher level of secreted IgA was observed in the bile and intestine in response to the oral booster. Although this vaccine developed an immune response before hatching, its protective potential was not evaluated [37].

The first subunit vaccine experiments were conducted in chickens using flagellin, the immunodominant antigen of *Campylobacter*. Bacterial flagella are composed of this component, and they play a crucial role in colonization. There was no consistency in the results of subunit flagellin vaccinations between studies. Based on research conducted by [38], flagellin combined with the Montanide adjuvant reduced the number of colony-forming units by three log10 compared to a control group. According to Huang et al. [39], flagellin vaccination was tested using DNA delivered through the intranasal route using chitosan nanoparticles containing the flagellin A vector pCAGGS-flaA. A significantly higher titer of specific antibody titer was observed for both serum IgY and intestine mucosal IgA after the second and third immunizations compared to the control groups, as well as a 2–3 and 2 log10 CFU/g reduction in bacterial loads in the large intestine and caecum, respectively, after an oral challenge. At the end of the study, *C. jejuni* was not found in the small intestine, proving the effectiveness of the *C. flagellin* vaccine.

In subunit vaccine experiments, other antigens were tested. On day 1 or day 15 after hatching, the CjaA protein, which is part of an ABC transporter system, was inoculated. The specific IgY titers were significantly higher in both experimental groups than in the control group during both inoculation periods. After the birds were first vaccinated on day 15, a similar load of *Campylobacter* was found in both groups on day 21 and slightly higher on day 28 post-challenge but always significantly lower than infected controls. The CjaA protein can be used as an immunization agent. It was found that nanoparticles decreased *C. jejuni* colonization in chickens by using nanoparticles. To encapsulate outer membrane proteins (OMPs), poly (lactide-co-glycolide) nanoparticles (PLGAs) were used. Despite different doses, *Campylobacter* load was not significantly reduced when administered orally. Contrary to the control group, subcutaneous vaccination produced a higher level of immunity since the colonization level in the gut dropped below the detection threshold [40]. The DPS gene played a role in colonizing *C. jejuni* in its host and could potentially be used as a vaccine antigen in a study investigating its role in biofilm formation. Chickens were not protected from *C. jejuni* colonization when recombinant Dps subunits were subcutaneously administered [41]. A subunit vaccination experiment has recently tested antigens for CadF, FlpA, and CmeC proteins, which are involved in the adhesion of *Campylobacter* to poultry. They all enhanced the sera reactivity of vaccinated birds when they used the Montanide adjuvant. The caecal load was not significantly reduced in CadF- or CmeC-vaccinated groups, but FlpA immunization significantly reduced the cecal load by approximately 3 log10 CFU g^−1^. As a result of vaccination with the fusion CaDF-FlaA-FlpA protein as well as with a mixture of the full-length individual proteins, the intestinal flora was also reduced by approximately 3 log10 CFU g. The results of these studies suggest that some *Campylobacter* antigens may have immunogenic properties, but further research and replication are needed to confirm their efficacy as vaccines [41].

*Salmonella* is one of the main bacteria affecting commercial poultry and the other (after *Campylobacter*) is one of the most important zoonotic foodborne pathogens. Several bacteriophage products against *Salmonella* infections are available today. In 2019, results were presented before the use of salmonella phages on a large scale in poultry production systems. Multiple administration of bacteriophage compounds (SalmoFREE^®^, THESEO, Laval, Mayenne, France) in drinking water was safe and did not affect chicken behavior or their production parameters. At the end of the cycle (day 33), *Salmonella* was reduced to 0% in cloacal swabs. Another product, Bafasal^®^ (Proteon Pharmaceuticals, Łódź, Poland) is a feed additive for birds and is administered with drinking water. In trials and commercial use, Bafasal^®^ has been shown to have a strong effect on food safety, reducing salmonella levels by up to 200-fold, while also improving feed conversion rates and reducing mortality. The Bafasal^®^ has both prophylactic and post-infection interventional effects, while its administration does not require a waiting period for meat and eggs [41]. Another product, Biotector S1^®^ (C.J. Chell Jidang Research Institute of Biotechnology, Dongho-ro, Jung-Gu, Seoul, South Korea), is the world’s first bacteriophage product, which can be used as a feed additive to control *Salmonella* enterica subspecies in poultry. Mortality recorded in the three experimental groups of commercial broilers (5-week-old Ras) that received Biotector S1^®^ at different concentrations in the feed (5 × 10^7^, 1 × 10^8^, and 2 × 10^8^ PFU/kg), decreased by 73% compared to the control group (11.81%) after the challenge. There was no significant difference in mortality between the experimental groups (2.78%, 3.13%, 3.13%). In the group of broiler breeders (67-week-old Ross) receiving bacteriophages (1 × 10^6^ PFU/kg), mortality (45%) decreased by 53% when compared to non-phage treatment controls (85%) after the challenge. The greatest reduction in mortality (by 86%) was seen in layers (6-week-old Lohmann Brown) who received the same dose (1 × 10^6^ PFU/kg) before the challenge. The mortality rate in the control group after the challenge was 35%. In the high-line brown layers, performance was also improved in the phage-treated group (1 × 10^8^ PFU/kg): a 3% increase in egg yield (90.6% in the trial, 87.5% in the control), and a 2.4% increase in egg mass (g/day/bird) (59.2% in the trial, 56.8% in the control) [42].

## 6. Bacteriophages and Food Safety/Security Regarding Poultry Meat

*Campylobacter* and *Salmonella* are the two most common infections connected with chicken. A comprehensive assessment of the former is available in [42]. Using phage cocktails, Chinivasagam et al. [43] showed that campylobacter could be controlled in broiler chickens from the farm to the processing facility with phage cocktails. The phage cocktails targeted *C. jejuni* and *E. coli*. To prepare these phages, 47-day-old birds were given them for 24 h before slaughter. Broilers ready for the market generally had lower *Campylobacter* levels after receiving the cocktails. As a result of high cecal *Campylobacter* counts and low phage titers in a few birds, the authors recommended a treatment time of 24 h for biocontrol of *Campylobacter in vivo* as per Chinivasgam et al. [43]. Researchers found that two phage cocktails could kill *C. jejuni* in broiler chickens, according to Richards et al., 2019. After two days of therapy, the caecal levels of the bacterium were significantly reduced (2.4 log10 CFU/g). Researchers found that after phage delivery with predation on *C. jejuni* in broiler chickens, the microbiota of the chicks remained unaffected, unlike antibiotics’ broad bactericidal effects. The research also reveals the *Campylobacter* control issue while reassuring the reader that phages are selective for their target bacteria, according to Richards et al. [18]. It is still possible for phage products to be used to control *Campylobacter* commercially. This was investigated in a broiler study using a phage cocktail containing three lytic phages [44] and how the timing of therapy affected the outcome. Infecting the chicks on hatching day with *S. enteritidis* was the first step in the infection process. The phage treatment was given early after bacterial infection (6–10 days) and late after bacterial infection (31–35 days).

In both in vivo trials, *S. enteritidis* counts were significantly reduced, but the late phage application was more effective. In poultry, multiple treatments were shown to be more effective at controlling intestinal *S. enteritidis* colonization with a reduction of 1.08 log CFU/g (from a starting concentration of 4.44 log10 CFU/g) [44]. Using SalmoFREE1, a previously patented phage preparation for fighting *Salmonella*, [45] investigated its efficacy. Phage products were fed to broiler chickens via their drinking water three times during their production cycle on a commercial farm. After and before treatment with the phage product, cloacal swabs were found to be effectively suppressing *Salmonella*. *Salmonella* counts in controlled broiler houses where the bacterium was still present on day 34 were 0% as compared with 0% in treated broiler houses. Using phages in poultry is showing great promise based on the research that has been conducted so far.

## 7. Biocontrol of Phages in Food Processing

With consumers becoming more aware of chemical food preservatives’ potential negative effects, they are increasingly preferring naturally produced foods that are minimally processed, chemical-free, and still safe to consume. These demands raise safety concerns, which make it difficult to satisfy them. The use of phages in this scenario can provide a natural means of eliminating harmful bacteria without harming humans or animals. There is no shortage of them in the environment, they occur naturally in food and water, and they play a significant role in the human microbiome [46]. Food products in the U.S. and European Union can be controlled with phage preparations approved by the FDA to control some of the most prevalent bacterial diseases. Several studies in the literature have documented how phages have improved food security naturally [47].

## 8. Bacteriophages as Antiseptic and Disinfectants in the Poultry Industry

### 8.1. Role as Antiseptics

Among poultry industry concerns are bacterial and fungal infections that occur in the environment of the birds. There is an increase in recovery potential on carcasses entering the processing plant when pathogenic bacteria are prevalent in poultry houses. Humans can become infected with *Salmonella* and *Campylobacter* through poultry products that have been cross contaminated in the kitchen, inadequately cooked, or handled improperly. Due to its ability to threaten consumer markets and increase production and processing costs, *Salmonella* adversely impacts the chicken industry [48]. *Salmonella* and *Campylobacter* can spread from feed mills to primary breeders, hatcheries, grow-out farms, processing plants, and finally to the final product because of poultry industry manufacturing procedures. It is common for *Salmonella* to be present in the environment, and there are several opportunities for birds to be exposed to *Salmonella* and other diseases during their growth period [49]. It is possible for these infections to persist in the environment for a long time and are usually found in bird litter. Disease populations can increase in the bird’s surroundings when feces are present in the litter. The hygiene of poultry houses is critical in the control and prevention of harmful infectious illnesses. An effective sanitation program can help growers by improving bird performance and reducing the prevalence of infected flocks. A sanitation program can include any number of best management practices, treatments, or disinfectants. However, if sanitation measures are not applied effectively, they might have a negative impact on disease prevention, decreasing bird populations. performance [48]. As a result, it is essential to regularly assess the performance of chicken house cleanliness procedures. Cleaning and disinfecting facilities after removing old litter can help to reduce pathogen loads and break disease cycles. Cleaning and disinfection will usually take place after the fifth or sixth flock cycle. The purpose of disinfecting any inanimate object, surface, or premises is to decrease or eliminate microbial populations [50]. Since most chemical disinfectants are limited in their effectiveness when organic matter and soil are present, greater sanitation programs require that bed materials be removed from the facility. There are several methods for applying disinfectants, such as spraying, fogging, fogging, and fumigation [50]. A variety of pesticides are available, including alcohol, halogens, quaternary ammonium compounds, phenolics, aldehydes, and oxidizing agents (such as ozone). The effectiveness of antimicrobial drugs is often determined by testing them against suspensions of laboratory bacteria. It may be difficult, however, to determine the true effectiveness of a disinfectant by using this method since it does not always simulate commercial production conditions. In some cases, disinfectants that work against bacterial suspensions may not be as effective against bacteria that are acting on surfaces [51].

### 8.2. Role as Disinfectants

The quantity of germs found in poultry operations has been reduced using a variety of methods, including strong biosecurity measures, regulatory restrictions, and specialized circumstances. Given that the flock was identified as the primary cause of contamination in chicken meat, it stands to reason that decreasing flock frequency would result in meat that is without germs. Additionally, spraying poultry and litter with an aerosol in production facilities may aid in preventing horizontal disease spread. Materials developed on bacteriophages can be utilized as sanitizers in hatcheries, farms, shipping containers, avian industrial plants, and areas that come into touch with food. Viral vectors are also thought to be effective at preventing the development of complete infections caused by harmful bacteria on surfaces that are frequently seen in the chicken industry. The composition of the components of apparatus matter a lot. *Salmonella* spp. were observed to develop biofilms on glass and stainless-steel surfaces substantially more than they did on surfaces made of polyvinyl chloride (PVC). Additionally, compared to other studied serovars, *S. enteritidis* and *S. Heidelberg* demonstrated a better capacity to build biofilms [52]. Some products are being used in developed countries to avoid bacterial contamination of the products. As an example, BacWash^TM^ (OmniLytics Inc., Sandy, UT, USA) has been effectively used as a spray (sprinkle, wash, or splatter) on live animals before sacrificing/slaughtering animals. Similarly, Ecolicide PX (Intralytix, Inc. Headquarters, Columbia, MD, USA) *E. coli* O157:H7 was formulated to clear the animal’s skin before slaughtering [53]. According to Atterbury et al. [17], bacteriophages can decrease the quantity of detectable *C. jejuni* cells on broiler chicken skin that has been experimentally polluted. Five phages from chicken feces that were described and chosen for usage as biosanitizers were evaluated by additional authors [54]. To reduce *Salmonella* Enteritidis on chicken skin, researchers examined the effectiveness of a phage cocktail and chemical agents. Skin tissues were deliberately infected with 1 × 10^5^ CFU/cm^2^ *Salmonella* by dipping them in 100 mL of the phage cocktail at 10^9^ PFU/mL for 30 min. Researchers have shown that retroviruses with short contact times and chilling temperatures decreased *S. enteritidis* burdens on chicken skin. After applying active ingredients, a comparable amount of *Salmonella* reduction (by an average of 1 log CFU/cm^2^) was also attained. El-Gohary et al. [55] revealed that treating garbage with a bacteriophage product that targets E. coli was an effective and feasible method of preventing colibacillosis in broiler chickens brought on by environmental infection to *E. coli*.

### 8.3. Limitations

Bacteriophages may stay longer on the surface of processed meat due to refrigeration conditions [56]. It would seem beneficial to enhance phage classification techniques and keep them isolated from the host environment to attain their highest levels of efficacy [57]. Finding new methods for bacteriophage purifying that ensure the proper elimination of components of bacterial origin is crucial. Phages must not be allergic or pyrogenic (sterility test, lack of residual endotoxin). The fact that bacteriophages may produce antibody humoral reactions and that they are immunogenic is commonly acknowledged to have an impact on phage treatment in both humans and other species, especially chickens [58]. Even though the initial oral safety trial in humans revealed no anti-phage antibodies [59]. The results of treatment may not always be affected by the antibodies that phage therapy induces, according to later investigations. Additional findings indicate that the method of phage delivery and phage type affect the antiphase activity in human sera [60]. However, various phages could react differently to antibody neutralization. It is unknown how the immune system of the host and the phage interact. Some writers have suggested testing phages for their capacity to prevent antibody neutralization [61]. It was reported (ECDC, 2019) that campylobacteriosis, salmonellosis, Shiga toxin-producing *E. coli* (STEC) infection and yersiniosis were the most common zoonoses in the European Union [62].

## 9. Reduction of Food Contaminations

Decontamination is the process of removing and neutralizing (inactivating) germs from food products to delay the putrefactive and aging processes. Eliminating bacteria from food is a key component of fighting human diseases. There are numerous techniques used to increase food safety, but not all of them are suitable for fresh meats and goods, and their application is restricted. Some examples, including heat pasteurization, high pressure, and radiation, are considered. Although with other techniques, when we use organic acids that are efficient (because of their low pH), the killing of the bacteria that was treated necessitates a significant amount of acid, which could affect the meat’s quality and appearance [9]. Phages have been used to reduce infections in raw and raw meat (RTE) foods as well as to decontaminate corpses [57]. Phages are now used as food biopreservatives in several commercially available products that fight *Listeria monocytogenes*, *Salmonella*, Shigella, and *E. coli*. Due to its capacity to thrive and reproduce at refrigerated temperatures, *Listeria monocytogenes* offers a unique risk for foodborne illness. From 2014 to 2018, there has been a rise in the number of confirmed cases of human listeriosis [62]. ListShieldTM (formerly LMP-102, Intralytix Inc., Baltimore, MD, USA) has received approval from both the United States Food and Drug Administration (U.S. FDA) and the United States Department of Agriculture (USDA) for use as a food additive to eat products that are made up from the poultry and beef. Specifically aimed at *L. monocytogenes*, ListShieldTM is a concoction of six lytic bacteriophages that can be used on food. ListShieldTM’s maker asserts that it has no impact on the organoleptic quality of food [63].

## 10. Adverse Effects of Bacteriophage in Poultry

Most importantly, every phage-based product meant for use in poultry veterinary medicine, poultry production, and the poultry industry should be secure and efficient. The timing of phage-based product administration, including dosage, delivery method, and concurrent use of medication, is most important [64]. Each bacteriophage’s ability to stay in or on food may differ, as may the application conditions and temperature. Bacteriophage persistence on the surface of meat products may be improved by refrigeration temperatures. Controversies and debates have arisen because of the clearance to use bacteriophage as directly apply to food. Even though there has been a lot of research on bacteriophage applications and numerous conclusive findings, there are still some drawbacks and unanswered questions [65]. Furthermore, the limited spectrum of phage activity may provide difficulties in the control of disease and may limit the sorts of infections for which such a strategy may be effective [22]. The genetic material from lysogenic phages is incorporated into the bacterial genome. As a result, they might serve as carriers for horizontal gene transfer between microbes, animals, or humans through the food chain. It now appears possible to comprehend how genes move between phages and their hosts because of study advancements. As a result, it would also be possible to prevent potentially harmful bacteriophages or to redesign them so that they are not capable of disseminating any other types of genes or unwanted features. Numerous writers combined various phages into one product, broadening the lytic spectrum and postponing the development of phage resistance. The effectiveness of bacteriophages depends, among other things, on their ability to replicate well and endure in specific environments. It is advisable to strengthen phage selection techniques and isolate them from the host environment to reach their maximum efficiency [57]. Finding new methods for phage purification that ensure the proper elimination of components of bacterial origin is crucial. Phages must not be allergic or apyrogenic (sterility test, lack of residual endotoxin). The fact that bacteriophages can produce specific antibody humoral responses and are immunogenic is commonly acknowledged to have an impact on phage therapy in both humans and other species, including chickens [58]. Even though the initial oral safety trial in humans revealed no anti-phage antibodies [59], the results of therapy may not always be affected by the antibodies that phage therapy induces, according to later investigations [66]. Other findings indicate that the way of food delivery and the phage can affect the antiphage activity in human sera [67]. Phage inactivation is not always the result of interactions between the phage and the antibody. However, different phages might react differently to antibody neutralization. A phage’s immune system interacts with the immune system of its host but how is unknown. Testing the capacity of phage-types can prevent neutralization of antibodies [68]. Modern technical techniques have been created to maximize the activity of phages that treat the inner cells’ illness and the covering and protection of those that are given to animal food. One such technology is phage encapsulation [69].

## 11. Challenges in the Application of Bacteriophages in Poultry

To combat the emergence of multi- and pan-drug resistant bacteria, new antimicrobial agents and therapeutic strategies are needed. Bacteriophages are one of the most viable remedies to the current medical crisis, offering numerous advantages over conventional antibiotics as we move from the antibiotic era to the post-antibiotic era. It is evident that phage therapy (PT) has made significant progress over the past two decades, but gaps still need to be identified and filled, the experience needs to be a complement, and further evidence of the efficacy and safety of PT must be provided, as well as reconsidering methods and practical approaches to achieve beneficial results for human health and wellbeing [56]. Preparations based on phages must be safe and effective for use in poultry production, poultry medicine, and the poultry industry. Besides dosage and route of administration (including preparation of standardized formulations), timing and concomitant preparations (such as competitive exclusion) or vaccinations should also be considered when administering phage-based products. The persistence of bacteriophages in/on food can be affected by bacteriophages themselves as well as environmental factors (such as temperature). At refrigeration temperatures, bacteria may persist longer on meat products [70].

Phage preparations need to be safe for application, which at times presents manufacturing and formulation challenges. The production of phages on a large scale under Good Manufacturing Practices (GMPs) that have been approved by regulatory agencies is necessary before they can be used widely in medicine [71]. Pharmaceutics must comply with strict regulations to ensure high-quality standards appropriate for the use of phages for therapy. Despite this, no specific guidelines for the production of phages have been developed [72]. To address this issue, researchers have developed quality and safety criteria for phage therapy products [73]. Phages that encode lysogeny, virulence genes, or antibiotic-resistant bacteria must be avoided to prevent the spread of these factors. Phage therapy may be ineffective against some fastidious bacteria, such as *Clostridium difficile*, for which there are no strictly virulent phages yet available [74]. Additionally, a threshold should be established to determine whether phage preparations contain impurities, such as endotoxins [75]. Several purification methods have been developed to remove these toxic elements from phage preparations [76], but none have yet achieved optimal results.

Producing phages in large quantities to meet the poultry market’s needs may be economically challenging. Using a bioprocess model and economic analysis, Torres-Acosta et al., [77] studied the scaled-up generation bacteriophage cocktail intended for poultry farming. According to the results, the most cost-effective method was to use one bioreactor (156 L) for six phages, followed by a 0.45 m filtration to remove the biomass and a 0.22 m filtration to ensure sterility. In experimental-theoretical calculation, it has been shown that the configured system can produce 210 million chickens for $0.02 per chicken using the applied configuration. To reduce production costs, the production titer must be optimized and improved [78].

## 12. Future Perspective

### 12.1. Regulatory Authority

The role of phage therapy in phage ecology and evolution with particular emphasis on interactions between bacteria and phages is often overlooked. Quorum sensing (QS) modulates bacterial susceptibility to phages, while phages modulate bacterial cooperation. It is recommended to examine these interactions carefully to maximize the antibacterial properties of phages [79]. Few studies have examined whether PT (Phage therapy) can be used in low and middle-income countries. In the deployment of phage therapy, the World Health Organization (WHO) must play an important role, according to the authors’ recommendations. A vaccine prequalification program (PQ) by the WHO, for example, could contribute to the promotion of PT knowledge and the development of a regulatory system for phage products [80].

### 12.2. Combination Therapies

Combining phages with other agents may be more effective in controlling bacterial infections, especially when targeting complex biofilm communities [81]. Combination therapies reduce resistance towards agents with distinct modes of action [78,82] because of the fitness cost associated with multiple factors [82]. In addition to phages, enzymes can also be used in conjunction with them to improve their activity. It is possible to combine depolymerase with phages that do not naturally express them to increase their activity against biofilms. Besides modifying phage genomes, improving phage therapy outcomes is also being investigated. Synthetic biology techniques have now made it possible to engineer phage genomes, with several techniques available for doing so [83,84].

### 12.3. Host Range Interactions

The host range of phages is one of the most important targets for phage engineering. Phages are highly specific to hosts, so they cannot target beneficial bacteria, but they can only target a limited number of strains within a species. The future of PT will be personalized, suggesting that personalized therapies are possible within 25 years. A series of events led to the creation of a cell-free synthetic phage, starting with community efforts, and culminating with the implementation of Artificial Intelligence and Distributed Ledger Technology. Despite the feasibility of this ideal scenario in theory, [76] acknowledges that obstacles can disrupt it easily.

## 13. Conclusions

Antimicrobial resistance has become a source of reduction in production and an equal threat to public health. Bacteriophages have arisen as an effective treatment alternative to the use of antibiotics by eliminating pathogens resultant rise in meat production. Several products are used in several different formats, and a lot of research is required to formalize the general and economical use of these phages. Considerable economic losses have been reduced by applying this novel technique. There are, however, some limitations, such as adverse reactions, infectivity itself by a bacteriophage, eliminating beneficial bacteria, and dose standardization, that must be a source of focus.

## Figures and Tables

**Figure 1 animals-12-03378-f001:**
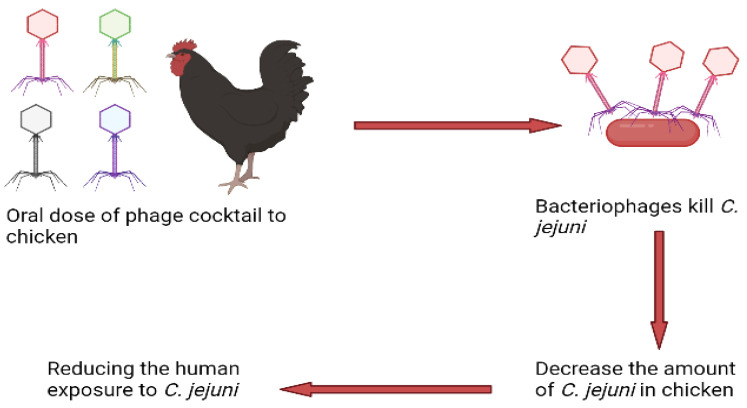
Mechanism of the phage cocktail to kill the *C. jejuni*. Orally administered phage cocktail to infected chicken. The phages kill the *C. jejuni* and decrease the level of *C. jejuni* in chickens. As a result, reduced human exposure and morbidity.

**Figure 2 animals-12-03378-f002:**
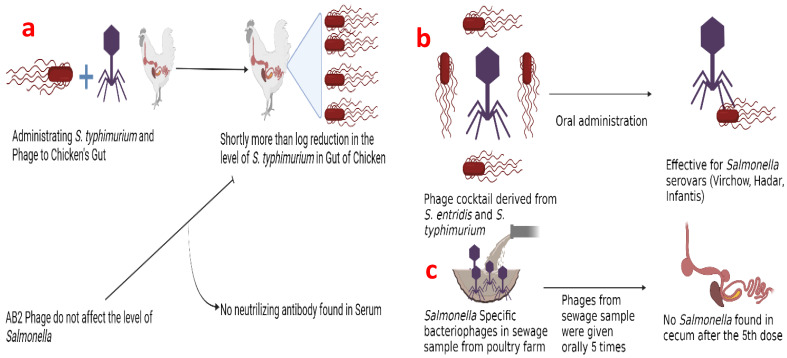
Shows how phages work against *Salmonella*. (**a**) delivering *S. typhimurium* and phages to the gut of the chicken. In no time, the level of *S. typhimurium* will reduce by Log_10._ While AB2 phages do not have any impact on the level of salmonella, thus no neutralizing antibodies are found in the serum. (**b**) Oral administration of phage cocktails derived from *S. enteritidis,* and *S. typhimurium* will be effective for salmonella serovars (Virchow, Hadar, Infentis). (**c**) *Salmonella*-specific bacteriophages in a sewage sample from a poultry farm given orally 5 times. As a result, no salmonella was found in the serum.

**Figure 3 animals-12-03378-f003:**
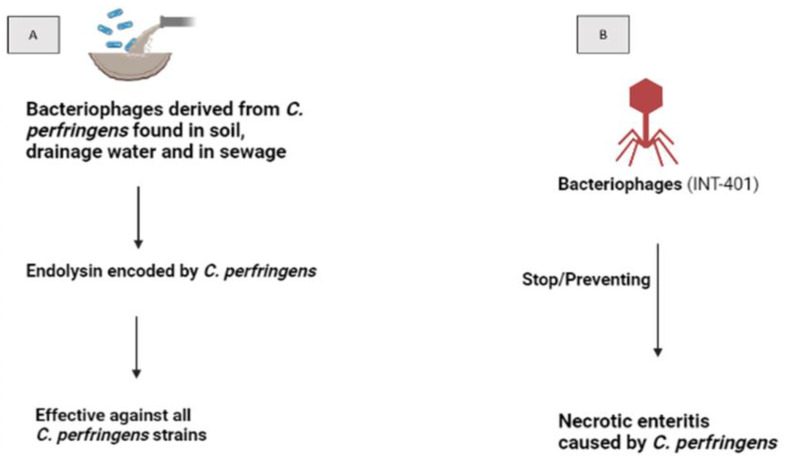
(**A**) *C. perfringens* bacteriophages found in soil, drainage water, and in sewage. Phages produced endolysin which causes cell breakdown and release the newly formed virions. Effective against all *C. perfringens*. (**B**) It has also been reported that bacteriophages (INT-401) prevent necrotic enteritis caused by *C. perfringens*.

**Figure 4 animals-12-03378-f004:**
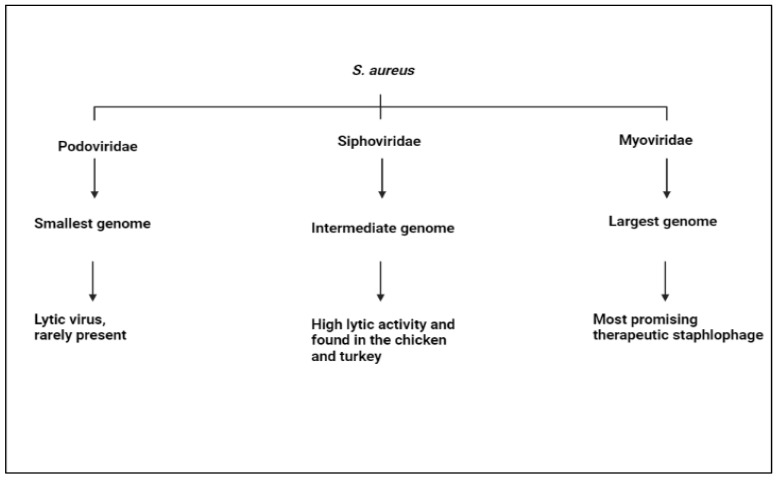
*S. aureus* has three staphylophages. Podoviridae has the smallest genome, lytic virus, but it is rarely present. Siphovidae has an intermediate genome and high lytic activity, and it is found in chicken and turkey. Myoviridae has the largest genome and is the most promising therapeutic staphylophage.

**Figure 5 animals-12-03378-f005:**
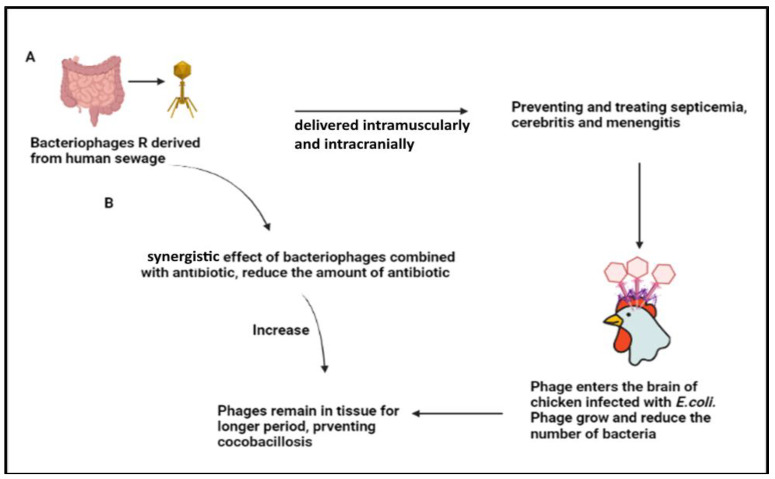
(**A**) Bacteriophages R derived from human sewage delivered intramuscularly and intracranially that prevent and treat septicemia, cerebritis, and meningitis. Phage enters the brain of infected chicken, reducing the number of bacteria, and remaining within the tissue for a longer period, thus preventing colibacillosis. (**B**) shows the synergistic effects of bacteriophages with an antibiotic, which increases the prevention of colibacillosis.

## Data Availability

Not applicable.

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
