# Peer review of "Role of Bacteriophages for Optimized Health and Production of Poultry"

_animals, 2022, doi:10.3390/ani12233378_

Round 1

Reviewer 1 Report

Overall analysis of manuscript: The manuscript is highly specific and need of the hour for poultry production and health optimization having sufficient knowledge of bacteriophages with perspective to poultry. It is highly suited in current scenario and expected to open new and effective 

General comments:

• Scientific names of organisms are not italic.

• Grammatical typos are seen in the manuscript

• Write history of bacteriophages in brief

• It is better if review of products of bacteriophages used in poultry is added in the manuscript

Specific Comments

1. Line 16-17: Sentence needs clarity

2. Line 19-20: This sentence needs to be revised

3. Line 31-32: Sentence is lacking clarity

4. Line 41-42: Please consider this sentence for proper understanding

5. Line 42-43: Please explain this sentence to highlight background

6. Line 48-49: Please revise it

7. Line 60-61 belongs to last paragraph where objective of the review is mentioned

8. Line 119-20: Please consider this sentence for proper understanding

9. What does mean by “Bacteriophage R” in line number 139

10. Line 144-45” Phages entered…. with E. coli” needs to be revised

11. Revise subheading “Bacteriophages in the enhancement of immunity of poultry”

12. Line 211-212 seems un-necessary, better to delete it

13. Line 222-223: Lower prevalence of what?

14. Subheading 4.4 S. aureus: It is better if staphylophage is mentioned in this section

15. Subheading 9 and subheading 13 seems repetition or there is typing error. Please check and correct

16. Subheadings 15 and 16 cover similar meaning. Can it be combined, or the subheadings are revised clearly distinct meanings?

Author Response

We are grateful to you for your valuable comments that helped us improve manuscript. Followings are responses to the comments. The revisions are yellow highlighted in the manuscript.

Overall analysis of manuscript: The manuscript is highly specific and need of the hour for poultry production and health optimization having sufficient knowledge of bacteriophages with perspective to poultry. It is highly suited in current scenario and expected to open new and effective 

General comments:

  • Scientific names of organisms are not italic.

Response: Correction has been made

  • Grammatical typos are seen in the manuscript

Response: Correction has been made as suggested

  • Write history of bacteriophages in brief

Response: Added as suggested

  • It is better if review of products of bacteriophages used in poultry is added in the manuscript

Response: Edited as suggested

Specific Comments

  1. Line 16-17: Sentence needs clarity
  2. Line 19-20: This sentence needs to be revised
  3. Line 31-32: Sentence is lacking clarity
  4. Line 41-42: Please consider this sentence for proper understanding
  5. Line 42-43: Please explain this sentence to highlight background
  6. Line 48-49: Please revise it
  7. Line 60-61 belongs to last paragraph where objective of the review is mentioned
  8. Line 119-20: Please consider this sentence for proper understanding
  9. What does mean by “Bacteriophage R” in line number 139

Response: This was typo which has been eliminated

  1. Line 144-45” Phages entered…. with E. coli” needs to be revised
  2. Revise subheading “Bacteriophages in the enhancement of immunity of poultry”
  3. Line 211-212 seems un-necessary, better to delete it

Response: Deleted as suggested

  1. Line 222-223: Lower prevalence of what?

Response: It was about campylobacter. Correction made as suggested

  1. Subheading 4.4 S. aureus: It is better if staphylophage is mentioned in this section

Response: Added as suggested

  1. Subheading 9 and subheading 13 seems repetition or there is typing error. Please check and correct

Response: Thank you for identification of this mistake. We have deleted 13 subheading and its material because it was repetition of subheading 09

  1. Subheadings 15 and 16 cover similar meaning. Can it be combined, or the subheadings are revised clearly distinct meanings?

Response: Thank you for identifying this overlapping meaning of subheadings. We have corrected subheadings. Moreover, the subheading “economics of bacteriophages in poultry” has also been merged with “challenges in application of bacteriophages”

Reviewer 2 Report

MANUSCRIPT 1928908

Common comments

1. Please check typos

2. References should be cross-matched in the document

3. Some sentence structures need to be improved

4. Data from recent articles should be added

5. Focused comments:

6. Title better to be refined 

7. Subheadings 2 and 3 should be revised as they reflect the similar meaning

8. Subheading 3 should be further divided into sub-subheadings

9. Subheading 4 should be revised and concise

10. Subheading 8 and 9 carry similar information

11. Revise subheading 10

12. Subheading 11 should be merged with previous information where the description of phages is mentioned

13. Subheading 13 seems a repetition

14. Subheading 15 seems like previously described in the manuscript

15. Future perspectives are better to be divided into further subheadings or point by point

16. Conclusion as an overall needs to be revised

Author Response

Dear reviewer, we are very much thankful for your comments. We have tried our level best to understand and address comments. Please find responses to the comments point by point.

. Please check typos

Response: Typos have been eliminated

  1. 2. References should be cross-matched in the document

Response: References has been checked

  1. Some sentence structures need to be improved

Response: Revision has been made regarding sentence structure as suggested

  1. Data from recent articles should be added

Response: Data has been updated as suggested

  1. Focused comments:
  2. Title better to be refined

Response: Tittle has been refined as “Role of bacteriophages for optimized health and production of poultry”

  1. Subheadings 2 and 3 should be revised as they reflect the similar meaning

Response: We have revised subheading 2 as “Biomedical application of bacteriophages”

  1. Subheading 3 should be further divided into sub-subheadings

Response: Subheadings have been added as suggested. Subheading 3 itself has also been revised as “Experimental studies on bacteriophages in poultry”

  1. Subheading 4 should be revised and concise

Response: Subheading 4 has been revised as suggested

  1. Subheadings 8 and 9 carry similar information

Response: Subheading 8 has been revised as “Bacteriophages as antiseptic in poultry industry”

  1. Revise subheading 10

Response: Subheading has been revised

  1. Subheading 11 should be merged with previous information where the description of phages is mentioned

Response: Correction has been made as suggested

  1. Subheading 13 seems a repetition

Response: Subheadings are revised

  1. Subheading 15 seems like previously described in the manuscript

Response: Subheadings are revised as suggested, and data has been shifted where required

  1. Future perspectives are better to be divided into further subheadings or point by point

Response: Sub-subheadings have been made as suggested

  1. Conclusion as an overall needs to be revised

Response: Conclusion has been revised as suggested

Reviewer 3 Report

Dear Authors,
The subject of the review is very interesting and worth sharing with the broad audience. However, the manuscript is very poorly prepared, lacking logical coherence, with many sentences appearing out of context and unclear statements that makes the text very hard to comprehend. On top of that, extensive editing of English language and style is strongly required.
Please refer to some of my comments in the attached PDF.

As of now, I suggest rejecting the manuscript in order to give the Authors more time to work on the review.

Kind regards,
Reviewer

Author Response

Dear reviewer, we appreciate your critical comments to improve our manuscript significantly. We have acknowledged your both comments given in the pdf file as well separately. We have extracted those comments and their responses have been prepared. Please find responses to the comments while changes in the manuscript are yellow highlighted.

Comments and Suggestions for Authors

Dear Authors,
The subject of the review is very interesting and worth sharing with the broad audience. However, the manuscript is very poorly prepared, lacking logical coherence, with many sentences appearing out of context and unclear statements that makes the text very hard to comprehend. On top of that, extensive editing of English language and style is strongly required.
Please refer to some of my comments in the attached PDF.

As of now, I suggest rejecting the manuscript to give the Authors more time to work on the review. We appreciate and respect your review outcome. As per our understanding and your comments in the text, we have revised manuscript for better clarity, understanding, and novelty. We are hopeful our revised version will be acceptable to you.

Finally, your comments and review report has significantly improved our manuscript. Thank you very much for your time.

animals-1928908-.pdf review

Kind regards,
Reviewer

Response: Thank you very much for keenly reviewing our manuscript. We are thankful for giving us opportunity to revise this manuscript.

Affiliation: Remove spacing

Response: Space has been removed

L11: of into for

Response: changed as suggested

L15: Meaning is unclear

Response: Sentence has been revised

L25: Meaning of obsessed is not clear

Response: The sentence has been revised as suggested

L32: Phrase has been found 3rd time in the paragraph, avoid such redundancy

Response: We apologize for this mistake. We have deleted it.

L46: Keep it simple, please reword into “Poultry pathogen”

Response: We have revised it as suggested.

L52-53: This sentece appears out of context. Some introducion to why a wide range of antimicrobials is required; The context here is understandable, yet once more lacking some introducion.

Response: We have revised sentence for better clarity. Thank you for this comment

L55-56: This sentence should come first to keep the text in logical coherence.

Response: We have placed this sentence before reference 3 sentence

L61: "in poultry production"

Response: We have corrected it as suggested

L62,63, 64: Please provide reference: Meaning unclear, please reword: Meaning unclear.

Response:

Response: We have revised sentences for better clarity and deleted portion that seem ambiguous.

L68: This sentence seems very specific for an introducion.

Response: We have deleted this sentence to avoid any misleading fact

L76: Meaning unclear

Response: We have refined paragraph

L81: Redundance

Response: Deleted as suggested

Line 84: Redundace with lines 76-77

Response: Deleted as suggested

L85-86: This sentence is out of context at the same time being too specific for an introducion.

Response: Sentence has been deleted and paragraph has been refined

L87: Subheading 2: Meaning unclear, please reword.

Response: We have revised subheading for better understanding

Line 736: This looks like a draft section and I believe it should be removed, together with the items below, that are not applicable.

Response: We have deleted as suggested

Reviewer 4 Report

The subject of the manuscript is very interesting, but the organizational side requires a lot of work to be done. Below are my comments to the article:

1. Dear Authors, when writing Latin names of microorganisms, italics must be used. It is used selectively in this text - please correct it.

2. This is a scientific article, so do not use phrases like "Poultry production is obsessed ... (line 25)" as it is not an exact science formulation.

3. The quoted data on world production of poultry meat is from 2017, so I think that it should be newer, 5 years have passed - please update this data.

4. From the fragment of the text contained in lines 44-60 it is very difficult to read who / what this information refers to - people or animals? Please re-edit this passage so that it is understood unambiguously.

5. The text on lines 82-84 is very general and should be placed at the beginning of this chapter.

6. The signature for figure 2 is unusual - please edit it.

7. Figure 3 for Staphylococcus aureus is referenced in the text for Escherichia coli. Is it correct?

8. Chapter 4 should be logically connected to Chapter 3 as the messages complement each other.

9. In line 454 the authors write: "Accordingly, a study was conducted to determine if four commonly used poultry house disinfectants were effective in lowering total aerobic bacterial, yeast and mold, Campylobacter, and Salmonella populations in poultry houses and in sterilized topsoil" i this is where the paragraph/chapter ends. I was expecting an indication of the results obtained in this study, but I have not obtained such information.

10. Are BacWashTm (line474) and Ecolicide PX (line 476) preparations universally approved for use? Is it only by some institutions (e.g. in the USA)?

11. Please check the spelling of the units on line 482 - they are not written in superscripts, appropriate for notation of power values.

12. I believe that chapter 9 should be merged with chapter 13.

13. Chapter 14 seems redundant in this manuscript.

14. Currently, there are 88 items in the reference list, while only 87 in the text of the manuscript - one is missing.

15. The record of the cited items requires ordering and saving in accordance with the requirements of the Editorial Board. Total chaos - full names and abbreviations, sometimes there is a doi number, sometimes there is none. To improve!

Author Response

Dear reviewer, your comments are very much appreciable and have significantly improved manuscript. Please find responses of your comments for your reference.

The subject of the manuscript is very interesting, but the organizational side requires a lot of work to be done. Below are my comments to the article:

Response: Thank you for let us improve manuscript. We have revised manuscript as per your suggestion and feel that it shall be acceptable to you.

  1. Dear Authors, when writing Latin names of microorganisms, italics must be used. It is used selectively in this text - please correct it.

Response: Thank you so much for this comment. We have corrected typos

  1. This is a scientific article, so do not use phrases like "Poultry production is obsessed ... (line 25)" as it is not an exact science formulation.

Response: we agree with reviewer and appreciate for this comment. We have corrected as suggested

  1. The quoted data on world production of poultry meat is from 2017, so I think that it should be newer, 5 years have passed - please update this data.

Response: We agree with reviewer. Correction has been made as suggested

  1. From the fragment of the text contained in lines 44-60 it is very difficult to read who / what this information refers to - people or animals? Please re-edit this passage so that it is understood unambiguously.

Response: We have revised paragraph as suggested.

  1. The text on lines 82-84 is very general and should be placed at the beginning of this chapter.

Response: The sentences have been deleted because of low coherence in the manuscript

  1. The signature for figure 2 is unusual - please edit it.

Response: We have edited it, however, good quality figure shall also be provided on acceptance of manuscript. While pasting in manuscript, sometimes pixels are distorted.

  1. Figure 3 for Staphylococcus aureus is referenced in the text for Escherichia coli.Is it correct?

Response: No this was not correct. Thank you for the comment, we have correctly placed it under S. aureus subheading

8. Chapter 4 should be logically connected to Chapter 3 as the messages complement each other

Round 2

Reviewer 3 Report

The authors attempted to deliver a very interesting and important subject to the broader audience. Although the improvements to the introduction section after the first round of the reviews are significant, I still stand by initial decision to reject the manuscript in order to give the Authors more time to work on the review paper of this very comprehensive subject, which are bacteriophages.

The whole manuscript is very poorly structured, still lacking logical coherence. Similar information are scattered between different sections, resulting in a strong redundance of the information, with many sentenced used over again and often relating to the same, previously cited papers.

The manuscript covers a variety of topics related to the use of bacteriophages, from experimental studies, through therapeutical use to food safety. Unfortunately, this results in the vast amount of general and vague statements, without going into sufficient detail, and with considerable amount of cited papers being other review papers, rather than research articles. At the same time, many other subjects - that does not fit int the scope of manuscript - are attentively described, such as types and symptoms of bacterial infections (with further regard to the different bacteria strains), vaccinations with attenuated bacteria, or maintenance procedures at the poultry facilities.

I suggest the authors to work on the structure and coherence of the manuscript, starting with (1) general description of the problem (types of bacteria and brief description of the related infections), (2) introduction on why phages are a promising solution, (3) supporting the effectiveness of phages with specific research data, in one section instead of separate three redundant sections, (4) only then expanding to other areas (without further fragmentation to redundant sections!) such as (a) adverse effects, including influence on the poultry immunity (b) food security and, (c) human health.

Please find the attached PDF with comments highlighted in yellow or in red (if the text was already marked by authors in yellow).
